# Efficacy of Baduanjin Versus Brisk Walking on Cognitive and Physical Functions in Schizophrenia: A Three-Arm Randomized Controlled Trial

**DOI:** 10.3390/healthcare13233013

**Published:** 2025-11-21

**Authors:** Chyi-Rong Chen, Chien-Hui Chan, Tzu-Ting Chen, Yu-Chi Huang, Pao-Yen Lin, Liang-Jen Wang, Keh-Chung Lin

**Affiliations:** 1Department of Psychiatry, Kaohsiung Chang Gung Memorial Hospital and Chang Gung University College of Medicine, Kaohsiung 833401, Taiwan; ccr776@cgmh.org.tw (C.-R.C.); tracy5824@cgmh.org.tw (T.-T.C.); yuchihuang@cgmh.org.tw (Y.-C.H.); py1029@cgmh.org.tw (P.-Y.L.); 2School of Occupational Therapy, College of Medicine, National Taiwan University, Taipei 100225, Taiwan; 3Department of Nursing, Kaohsiung Chang Gung Memorial Hospital and Chang Gung University College of Medicine, Kaohsiung 833401, Taiwan; a6852135@cgmh.org.tw; 4Department of Child and Adolescent Psychiatry, Kaohsiung Chang Gung Memorial Hospital and Chang Gung University College of Medicine, Kaohsiung 833401, Taiwan; anus78@cgmh.org.tw; 5Division of Occupational Therapy, Department of Physical Medicine and Rehabilitation, National Taiwan University Hospital, Taipei 100225, Taiwan; 6Department of Occupational Therapy, College of Medical Science and Technology, Chung Shan Medical University, Taichung 40201, Taiwan

**Keywords:** schizophrenia, cognitive function, Baduanjin, brisk walking, exercise, randomized controlled trial

## Abstract

**Highlights:**

**What are the main findings?**

**What are the implications of the main findings?**

**Abstract:**

**Background:** Cognitive and physical deficits are core features of schizophrenia. Although Baduanjin and brisk walking (BW) have shown promise as intervention strategies, comparative evidence with follow-up and considering maintenance is limited. **Objective:** This study compared the effects of Baduanjin, BW, and health education (HE) on cognitive and physical outcomes in schizophrenia and examined whether a maintenance program could sustain these effects. **Methods:** In this single-blind three-arm randomized controlled trial, 60 patients with schizophrenia were assigned to Baduanjin (*n* = 20), BW (*n* = 20), or HE (*n* = 20). Interventions were conducted three times weekly for 12 weeks, each lasting 60 min, followed by a four-week home-based maintenance program with brochures and short message reminders. Cognitive outcomes were assessed using the Brief Assessment of Cognition in Schizophrenia, and physical outcomes included the Six-Minute Walk Test (6MWT), 30-Second Chair Stand Test (30CST), Timed Up-and-Go (TUG), motor dual-task TUG (TUGmanual), and cognitive dual-task TUG (TUGcognitive). **Results:** Baduanjin produced larger improvements than HE in verbal memory, attention and processing speed, executive function, and global cognition. BW significantly enhanced the working memory and global cognition versus HE, with additional improvements in attention and processing speed at follow-up. Both Baduanjin and BW improved the walking distance and lower-limb strength compared with HE, while Baduanjin outperformed BW and HE in balance and dual-task outcomes. **Conclusions:** Baduanjin and BW improved cognitive and physical functions in individuals with schizophrenia. Maintenance programs with short message reminders may help sustain these benefits.

## 1. Introduction

Cognitive impairment is a core symptom of schizophrenia and is closely associated with functional outcomes [1,2]. Individuals with schizophrenia also exhibit higher rates of metabolic syndrome [3], obesity, and cardiovascular problems [4]. Additionally, recent treatment guidelines have emphasized the importance of identifying effective interventions to enhance physical functioning in individuals with severe mental illness [5]. Antipsychotics are effective for positive symptoms [6] but have limited impact on negative symptoms [7] and cognitive function [8,9] in schizophrenia. Non-pharmacological interventions have become one of the most widely explored approaches in recent years.

Regarding cognitive function, treatment guidelines recommend physical exercise as a potential strategy for enhancing cognitive function in this population [10]. Recent meta-analyses have shown that aerobic exercise yields small-to-moderate improvements in global cognition [11]. Similarly, a meta-analysis of mindful exercises, including Baduanjin, yoga, and Tai Chi, reported comparable cognitive benefits in schizophrenia, with effects similar to those of aerobic or resistance training [12]. However, this study did not investigate specific domains of cognitive function (e.g., working memory, verbal fluency, etc.) [12]. The present study was extended to measure global cognition and domains of cognitive function in response to Baduanjin, brisk walking (BW), and control intervention.

Baduanjin is a traditional Chinese mindful exercise composed of eight distinct movements that emphasize coordinated motion, controlled breathing, and mindful awareness [13]. It aims to improve flexibility and muscle strength and is generally considered easier to practice than Tai Chi [14]. Baduanjin has been classified as a light-to-moderate-intensity aerobic exercise [15]; however, empirical studies examining its effects on psychiatric symptoms and functional outcomes in individuals with schizophrenia remain limited. Previous single-group studies have shown that Baduanjin training combined with short message reminders improved balance, cognitive function, and quality of life in individuals with severe mental illness [16]. A recent clinical trial reported that 12 weeks of Baduanjin training led to greater improvements in BMI and waist-to-hip ratio compared to controls [17]. A 24-week randomized controlled trial also demonstrated improvements in verbal memory [18]. Another 12-week randomized controlled trial found enhancements in overall cognition, balance, and dual-task performance among middle-aged individuals with schizophrenia. However, cognitive improvement was not retained at the 4-week follow-up assessment [19]. Of note, there was no intervention during the follow-up period [19]; therefore, there is a need for further investigation into a maintenance program during the follow-up period. To address this, the present study was designed to implement a maintenance program for the study groups during the follow-up period.

Walking is a feasible aerobic exercise for individuals with schizophrenia and has gained popularity in recent years. Previous studies have shown modest short-term reductions in BMI [20]. Clinical trials have reported improvements in cardiorespiratory function after 10-week walking interventions [21] and potential cognitive benefits following a one-year guided walking program [22], though both studies were limited by non-randomized designs. A randomized controlled trial also found that supervised moderate-to-high-intensity walking significantly improved attention and processing speed, but it did not include follow-up assessment [23]. Thus, the optimal walking regimen and its long-term effects remain unclear.

The maintenance program in the present study was designed in light of prior research on mental illness. Among individuals with severe mental illness, a post-intervention home-based exercise package comprising illustrated movement guides, instructional DVDs, and recording sheets, combined with twice-weekly short message reminders, yielded 81% adherence to the Baduanjin regimen [16]. More recent clinical trials show that text message reminders enhance exercise motivation in first-episode psychosis [24]. Systematic reviews further support the effectiveness of short message reminders in improving medication adherence and intervention outcomes in severe mental illness [25]. However, no randomized controlled trials have evaluated maintenance programs using text messages to sustain Baduanjin practice in individuals with schizophrenia.

To address the comparative efficacy gap in research into Baduanjin and BW in schizophrenia, the present study included a dose-matched control intervention and incorporated a more comprehensive repertoire of measurements for cognitive and physical functions. Our hypothesis posited that Baduanjin and BW would enhance cognitive and physical functions more than the control intervention (i.e., health education) immediately after treatment and at the 4-week follow-up assessment. In addition, we hypothesized that Baduanjin and BW would exert distinct effects on the outcome measures.

## 2. Materials and Methods

### 2.1. Study Design

This study was a single-blinded, three-arm, and parallel randomized controlled trial. Participants were allocated to the Baduanjin group, BW, or health education (HE) group. All participants were informed of the study purpose, and their consent was obtained prior to enrollment. Recruitment, intervention delivery, and follow-up assessments were conducted between January 2024 and January 2025 at the psychiatric day care center of a medical center in Taiwan. The study was approved by the Institutional Review Board and was registered on ClinicalTrials.gov (Identifier: NCT06226779, registered on 26 January 2024).

Following the baseline assessment, participants received their respective intervention for 12 weeks. A maintenance program was conducted to encourage participants to continue engaging in physical activities after the intervention for four weeks. Outcome evaluations were conducted at baseline, post-intervention, and at the 4-week follow-up.

### 2.2. Sample Size

The sample size estimation was based on the previously published literature, which reported a medium-to-large effect of Baduanjin on global cognition in patients with schizophrenia (η^2^ = 0.133) and a small-to-medium effect on lower extremity strength (η^2^ = 0.052) [19]. With a statistical power of 80% and an alpha level of 0.05, power analysis indicated that 9 to 18 participants per group were needed. Accounting for an anticipated dropout rate of 10%, we planned for a total of 65 participants. The sample size calculation was conducted using G*Power 3.1.9.4 software [26].

### 2.3. Participants

The inclusion criteria were as follows: (1) a diagnosis of schizophrenia according to DSM-5 [27]; (2) age between 20 and 65 years; (3) stable mental status with no consistent dosage for at least one month; and (4) ability to walk independently for 50 m. The exclusion criteria included (1) serious physical conditions such as cardiovascular, musculoskeletal, or cardiopulmonary disease; (2) visual or auditory impairments affecting assessment completion; (3) need for hospitalization; or (4) severe withdrawal or profound intellectual disability.

### 2.4. Randomization and Blinding

Eligible participants were randomly assigned to the Baduanjin, BW, or HE group in a 1:1:1 ratio using a web-based randomization program (www.randomizer.org: accessed on 1 Feburary 2024) managed by an independent research assistant who was not involved in the recruitment, assessment, or intervention. Outcome assessors were blinded to the group assignments, and participants were instructed not to disclose their intervention details.

### 2.5. Intervention

#### 2.5.1. The Baduanjin Group

Participants in the Baduanjin group practiced Baduanjin, while continuing standard medical and psychosocial care. Baduanjin exercise consists of eight postures, each emphasizing different body regions, performed in the following order: (1) Two Hands Hold Up the Heavens; (2) Drawing the Bow to Shoot the Eagle; (3) Separate Heaven and Earth; (4) Wise Owl Gazes Backward; (5) Sway the Head and Shake the Tail; (6) Two Hands Hold the Feet to Strengthen the Kidneys and Waist; (7) Clench the Fists and Glare Fiercely; and (8) Bouncing on the Toes. The practice protocol followed a previously published scheme [19]. Each session was led by a certified occupational therapist and conducted in small groups of six to eight participants. The sessions lasted 60 min and were held three times per week for 12 weeks. Each session included a 10-min warm-up consisting of range of motion and breathing exercises, 40 min of Baduanjin practice, and a 10-min cool-down with brief discussions of participants’ experiences and reflections. The attendance of the participants was recorded. After each session, participants rated their perceived exertion using a modified 10-point Borg scale [28].

After the 12-week program, participants were encouraged to continue practicing at home. To support adherence, a 4-week maintenance program was provided, including a picture-based Baduanjin brochure and text reminders. In accord with a previous protocol [16], two text reminders were sent weekly, with delivery days randomly assigned: one between 6:00 PM and 7:00 PM on a weekday and the other between 9:00 AM and 10:00 AM on the weekend. The amount of practice on the maintenance program was recorded.

#### 2.5.2. The BW Group

Participants in the BW group engaged in the activity for 60 min per session, three times per week for 12 weeks, while continuing their usual medical and psychosocial treatments. Each session consisted of a 10-min warm-up with range-of-motion exercises, 40 min of BW, and a 10-min cool-down with stretching. Sessions were conducted in groups of 6 to 8 participants and were supervised by occupational therapists who were independent from the Baduanjin group. The program targeted light to moderate intensity. Attendance was recorded, and after each session, participants rated their perceived exertion using a modified 10-point Borg scale [28].

After the 12-week intervention, participants were encouraged to continue the exercise practice at home. To support adherence, a 4-week maintenance program, including a picture-based walking brochure and short message reminders, was provided at the same frequency as in the Baduanjin group. The amount of practice at home was recorded.

#### 2.5.3. The HE Group

Participants in the HE group attended 60-min sessions three times per week for 12 weeks, covering health management topics and viewing videos unrelated to physical exertion, such as sports games or sports-related variety shows. Sports games and sports-related variety programs were selected because they are engaging, easy to follow, and do not elicit physical exertion, thereby serving as an attention-matched and expectancy-balanced control condition consistent with recommendations for active controls in behavioral and exercise trials [29]. Each session comprised a 10-min warm-up with discussion of daily life issues and HE content, 40 min of video viewing, and a 10-min facilitated discussion of viewing reflections. Sessions were conducted in groups of 6 to 8 participants and were led by a nurse. Attendance was recorded. After the 12-week intervention, participants received a home-based physical activity guide along with recommendations for video viewing and short message reminders.

### 2.6. Outcome Measurements

Baseline demographic and clinical data were retrospectively collected from medical records by the research team. These data included age, sex, educational level, marital status, living status, and duration of illness. The severity of illness, as measured by the Clinical Global Impression–Severity scale (CGI-S) [30], and the use of antipsychotic medications were extracted from medical records. Antipsychotic dosages were converted into chlorpromazine equivalents using the standard conversion formulas developed by Leucht et al. [31].

The primary outcome was the Brief Assessment of Cognition in Schizophrenia (BACS), which measures verbal memory, working memory, motor speed, verbal fluency, attention and processing speed, and executive function [32]. Global cognition was assessed by calculating a composite Z-score, reflecting the overall cognitive performance relative to a healthy control group [32,33]. The test–retest reliability of the Chinese version of the BACS has been established [34,35].

Secondary outcomes included the 30-Second Chair Stand Test (30CST) [36] and Six-Minute Walk Test (6MWT) [37], assessing lower-limb strength and cardiorespiratory fitness, respectively, with higher scores indicating better physical fitness. The Timed Up and Go Test (TUG) [38] along with the motor dual-task TUG (TUGmanual) and the cognitive dual-task TUG (TUGcognitive) were administered to evaluate balance and dual-task performance [39]. In the TUGmanual, participants performed the TUG while holding a cup filled to nine-tenths with water [39]. In the TUGcognitive, a number between 80 and 99 was randomly selected, and participants completed the TUG while performing serial subtraction by three [39]. For all TUG conditions, lower completion times indicated better functional performance.

### 2.7. Statistical Analysis

The demographic, baseline, and clinical characteristics of the study sample were summarized as the mean ± standard deviation for continuous variables and as the frequency (percentage) for categorical variables, as appropriate. Baseline comparability across the groups was assessed using one-way ANOVA for continuous variables and the χ^2^ test for categorical variables.

A two-way mixed analysis of covariance (ANCOVA) was performed to examine the effects of the three treatments on both primary and secondary outcomes, with baseline scores included as the covariate. ANCOVA is recommended for use in clinical research, because it adjusts for potential baseline differences, allows for valid group comparisons across all time points, and generally provides greater statistical power than alternative analytical approaches [40]. To assess within-group changes, paired *t*-tests with Bonferroni adjustment were applied to compare the baseline with the post-test and follow-up scores. Post hoc ANCOVA with Bonferroni correction was then conducted to examine differences among the three groups. Statistical significance was set at α = 0.05 (two-tailed), and the partial eta squared (*η*^2^) was calculated to estimate the effect sizes. According to Cohen’s guidelines, *η*^2^ values of 0.01, 0.06, and 0.14 correspond to small, medium, and large effects, respectively [41]. Statistical analyses were performed using the SPSS software version 25.0 (IBM Corp., Armonk, NY, USA).

## 3. Results

### 3.1. Background Information of Participants

Sixty-three participants were randomly allocated to the Baduanjin group (*n* = 21), the BW group (*n* = 21), or the HE group (*n* = 21). One participant in each group was excluded prior to the baseline assessment due to worsening psychiatric symptoms. The remaining 60 participants completed the study (Figure 1). There were no significant baseline differences between groups in demographic or clinical variables (Table 1). During the 12-week intervention, the mean attendance rates were 95.69 ± 0.04% in the Baduanjin group, 95.20 ± 0.04% in the BW group, and 94.44 ± 0.04% in the HE group. During this period, the mean ratings of perceived exertion were 4.67 ± 1.54 in the Baduanjin group and 5.02 ± 1.91 in the BW group. In the follow-up phase, the Baduanjin group reported a mean of 4.20 ± 1.39 home-based Baduanjin practice sessions per week, with a mean session duration of 21.20 ± 6.15 min. The BW group reported 5.10 ± 1.07 home-based walking sessions per week, with a mean session duration of 23.35 ± 5.33 min. No accidents or adverse events occurred during the study.

### 3.2. Primary Outcomes

Within-group analyses revealed that only the Baduanjin group showed significant improvements in verbal memory from baseline to both post-test and follow-up. In addition, significant gains in working memory, attention and processing speed, executive function, and global cognition were observed in both the Baduanjin and BW groups, whereas no such improvements were found in the HE group. There were no significant differences in primary outcomes between the post-test and follow-up assessments, suggesting that the intervention effects were maintained during the follow-up period (Table 2).

Repeated-measures ANCOVA revealed significant group effects in several cognitive domains of the BACS. Specifically, verbal memory showed a significant group effect (*p* < 0.01, *η*^2^ = 0.19), with the Baduanjin group performing significantly better than the HE group at both post-test and follow-up. Working memory also demonstrated a significant group effect (*p* < 0.01, *η*^2^ = 0.17), in which the BW group outperformed the HE group across post-test and follow-up assessments. For attention and processing speed (*p* < 0.01, *η*^2^ = 0.16), the Baduanjin group performed significantly better than the HE group at post-test, whereas at follow-up, both the Baduanjin and BW groups showed superior performance compared to the HE group. Notably, the executive function (*p* = 0.04, *η*^2^ = 0.11) showed a significant group effect, with the Baduanjin group demonstrating greater improvement than the HE group at follow-up. This suggests that the benefits of Baduanjin on executive functioning became more evident during the follow-up phase. In addition, both the Baduanjin and BW groups showed advantages in global cognition relative to the HE group (*p* < 0.01, *η*^2^ = 0.34) (Table 2).

### 3.3. Secondary Outcomes

Within-group analyses demonstrated that participants in both the Baduanjin and BW groups exhibited significant improvements from baseline to post-test and follow-up in 6MWT, 30CST, and TUG. Moreover, the Baduanjin group showed consistent within-group gains in TUGmanual and TUGcognitive across both time points. In contrast, the HE group showed minimal or no significant changes in any of the physical fitness measures. There were no significant differences in secondary outcomes between the post-test and follow-up assessments, indicating that the intervention effects were sustained over the follow-up period (Table 3).

The analysis suggested significant group effects across the physical fitness measures. For the 6MWT (*p* < 0.01, *η*^2^ = 0.22), the BW outperformed HE at post-test, whereas both the Baduanjin and BW showed superior walking distances compared with HE at follow-up. The 30CST (*p* < 0.01, *η*^2^ = 0.23) similarly revealed that both Baduanjin and BW demonstrated greater lower-limb strength than HE at post-test, and these benefits were maintained at follow-up. In terms of functional mobility, the standard TUG (*p* < 0.01, *η*^2^ = 0.33) showed that Baduanjin had faster completion times than both BW and HE at post-test and follow-up. In dual-task performance assessments, the Baduanjin group consistently demonstrated superior outcomes compared to both the BW and HE groups on the TUG-manual (*p* < 0.01, *η*^2^ = 0.29) and TUG-cognitive (*p* = 0.001, *η*^2^ = 0.34) tests, across both post-test and follow-up evaluations (Table 3).

## 4. Discussion

This three-arm randomized controlled trial evaluated the effects of Baduanjin, BW, and HE, along with a maintenance phase, on the cognitive and physical fitness outcomes in individuals with schizophrenia. The results suggest significant benefits of both Baduanjin and BW over HE in multiple cognitive domains and physical fitness outcomes. Specifically, individuals practicing Baduanjin showed greater improvements in verbal memory, attention and processing speed, executive function, and global cognition compared with HE, while BW was particularly effective in enhancing working memory and global cognition. In terms of physical fitness, both Baduanjin and BW improved the walking capacity and lower-limb strength relative to HE, whereas Baduanjin demonstrated additional benefits in balance and dual-task performance. Importantly, the advantages of Baduanjin and BW were not only evident at post-test but were largely maintained during the follow-up period, suggesting sustained effects attributable to the maintenance program. Collectively, these findings highlight the potential of Baduanjin as a mindful exercise and BW as an aerobic activity to enhance both cognitive and physical functioning in this population, with benefits sustained through the maintenance program.

The current findings show that Baduanjin was the only intervention to produce significant within-group improvements from baseline to follow-up and yielded higher gains in verbal memory compared with the HE group. Prior studies have reported promising memory-enhancing effects associated with Baduanjin [18], with a 24-week program involving 200 min of weekly practice significantly improving both the immediate and delayed logical memory on the Wechsler Memory Scale; however, no follow-up data were collected in that trial. Another study employing a shorter 12-week protocol with 120 min of weekly Baduanjin practice reported only short-term within-group improvements in logical memory among individuals with schizophrenia, with no sustained effects at follow-up and no advantage over BW [19]. Building on this evidence, the current study implemented a 12-week intervention involving 180 min of Baduanjin practice per week, followed by a maintenance phase with structured intervention strategies. The results revealed that Baduanjin produced significantly higher gains in verbal memory, as measured by the BACS, compared to HE. Moreover, these improvements were maintained at follow-up, highlighting the potential of Baduanjin as a sustainable intervention for enhancing verbal memory in this population.

With respect to attention and processing speed, earlier studies indicated that Baduanjin offered no clear advantage over BW [18,19]. In contrast, the present trial found that Baduanjin produced significant improvements relative to HE at post-test, and these benefits were maintained at follow-up. Additionally, while earlier studies found no significant effects of Baduanjin on executive function in individuals with schizophrenia [19], the current study revealed that Baduanjin led to significantly higher improvements compared to HE at follow-up. These benefits may be attributed to the longer weekly training duration and the incorporation of a structured maintenance phase. In present study, Baduanjin and BW showed similar perceived exertion ratings, indicating comparable intensity. This supports the interpretation that the higher benefits of Baduanjin stem from its mindful and balance-focused elements rather than intensity differences. Evidence also suggests that Baduanjin improves executive control through mechanisms beyond aerobic load [42]. Studies in older adults have shown that Baduanjin can influence the hippocampal subregion structure [43] and resting-state functional connectivity between the dorsolateral prefrontal cortex and the left putamen and insula [44], supporting its potential benefits in attentional regulation, memory formation, and cognitive control. Given that verbal memory, processing speed, and executive function are critical predictors of returning to work or school in individuals with first-episode schizophrenia [45], future research may explore the potential benefits of Baduanjin exercise in this population.

Our study also revealed that BW, as compared with HE, significantly improved working memory and global cognition at post-intervention. These benefits were sustained at follow-up, with additional gains observed in attention and processing speed. Recent meta-analyses have confirmed the efficacy of aerobic exercise in improving the working memory and overall cognitive function in schizophrenia [11], and trials focusing specifically on BW have also reported benefits in processing speed [23]. Prior research has additionally noted that BW is the most commonly adopted form of physical activity among individuals with schizophrenia [46], supporting its potential for long-term adherence. Collectively, our findings highlight that both BW and Baduanjin are effective strategies for promoting cognitive improvement in this population.

Neither Baduanjin nor BW produced significant changes in the BACS motor speed. Psychomotor slowing is common in schizophrenia and is closely related to functional performance [47], and recent studies likewise report the limited effects of exercise or cognitive training on this domain [23,48]. One possible explanation is that the BACS motor speed subtest requires participants to place coins into a bowl as quickly as possible, emphasizing rapid upper-limb movement [31], whereas Baduanjin and brisk walking do not specifically target upper-limb movement speed. Future research may examine whether activities involving fast upper-limb coordination, such as ball games, yield greater improvements on this measure.

Our findings indicate that both Baduanjin and BW produced significant and large improvements in global cognition compared with HE. This aligns with recent meta-analytic results, which suggest that mindfulness exercises can benefit cognitive function compared to treatment as usual, but they are not superior to other forms of exercise in individuals with schizophrenia [12]. Previous studies have shown that Baduanjin produces significant within-group improvements in global cognition following intervention [19]; however, these effects were not sustained at follow-up. Extending prior research, the present study incorporated short message reminders and picture-based brochures as part of a maintenance program. Short message reminders have been used to promote continued engagement in physical activity [49]. Here, a maintenance program with text reminders helped encourage participants to continue exercising at home. Participants reported practicing approximately four to five times per week for about 20 min per session, which aligns with prior findings and suggests that the interventions were acceptable and feasible for this population [16]. Such sustained engagement may have contributed to maintaining the intervention benefits and highlights the value of targeted follow-up strategies to support ongoing physical activity in individuals with schizophrenia.

Both the Baduanjin and BW groups showed significant post-intervention improvements in lower-limb strength compared to the HE group. The BW group demonstrated immediate improvements in walking distance, while the Baduanjin group showed significant gains at follow-up. These findings align with previous randomized controlled trials showing that both Baduanjin and BW yield comparable improvements in lower-limb strength and walking distance in individuals with schizophrenia [19]. A recent meta-analysis also revealed that traditional Chinese exercises such as Tai Chi and Baduanjin provide physical benefits similar to aerobic training [50]. Notably, the three-arm design of the current study further underscores the advantages of Baduanjin over HE in improving physical function.

In our study, Baduanjin produced larger and sustained improvements in balance than both brisk walking and health education, with a large effect size. This effect may be attributable to the emphasis on balance control in Baduanjin movements, including weight shifting and trunk rotation. This is consistent with previous research indicating that Baduanjin enhances balance in individuals with psychiatric disorders, including schizophrenia [16,19]. However, previous studies found that Baduanjin did not outperform BW in dual-task motor performance following the intervention [19]. In contrast, the present study employed a longer weekly Baduanjin practice than earlier trials, resulting in significant post-intervention improvements in both motor and cognitive dual-task performance, which were maintained through the follow-up phase.

This study has several limitations. First, all participants were recruited from a single institution and were limited to inpatients at a daycare center, which may limit the generalizability of the findings. Second, the small sample size may have reduced the statistical power for certain outcomes. Third, participants were not blinded to group allocation, introducing potential performance bias. Although assessors were blinded, participants and interventionists knew their group assignments, which may have introduced performance or expectancy bias, a common limitation in behavioral interventions [51]. The active control condition of health education and video viewing helped minimize this bias by providing comparable attention, social interaction, and session time across groups; however, participants’ expectations of greater benefit in the exercise groups could not be completely eliminated. Fourth, psychiatric symptoms were not assessed, leaving the effects on symptom severity uncertain. Although standardized symptom scales were not administered, overall clinical stability was monitored through the CGI-S, and the scores remained stable in both the Baduanjin and brisk walking groups. This stability suggests that the interventions were well-tolerated and supports the safety of incorporating structured exercise into rehabilitation for individuals with schizophrenia. Lastly, biomedical and neurophysiological measures were not included. Future studies should recruit participants from multiple sites, include larger sample sizes, enroll individuals at different stages of schizophrenia, and incorporate biomarker assessments to strengthen the validity and generalizability.

## 5. Conclusions

This study showed that both Baduanjin and BW improved cognitive functions, lower-limb strength, and walking distance in the participants with schizophrenia. The maintenance program was feasible for encouraging continued exercise practice at home. Baduanjin exhibited superior effects on balance and dual-task performance compared to the other groups. The findings indicated the benefits of Baduanjin as a promising form of aerobic exercise that may be integrated into psychiatric interventions in schizophrenia. Further research based on multicenter trials is needed to validate the findings.

## Figures and Tables

**Figure 1 healthcare-13-03013-f001:**
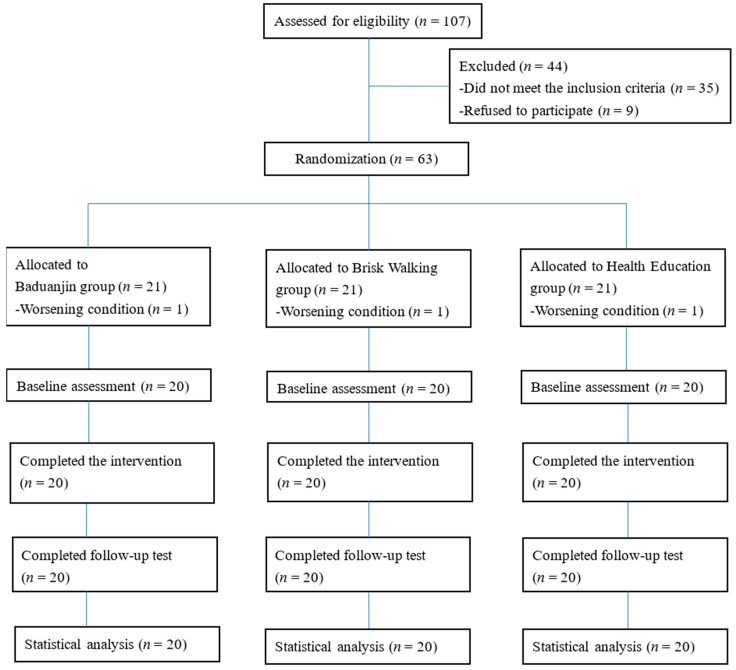
Participant flow diagram of the study.

**Table 1 healthcare-13-03013-t001:** Demographic and clinical characteristics of the study participants.

Variable	Baduanjin (*n* = 20)	Brisk Walking (*n* = 20)	Health Education (*n* = 20)		
	Mean (SD), Frequency (%)	Mean (SD), Frequency (%)	Mean (SD), Frequency (%)	*F*, χ^2^	*p*-Value
Age (years)	45.75 (10.96)	44.40 (12.58)	43.50 (12.06)	0.18	0.83
Sex					
Male	12 (60)	13 (65)	11 (55)		
Female	8 (40)	7 (35)	9 (45)	0.42	0.81
BMI (kg/m^2^)	25.37 (5.05)	26.91 (5.42)	27.56 (5.11)	0.94	0.40
Education					
Primary school	0 (0)	0 (0)	1 (5)		
Junior high school	5 (25)	3 (15)	6 (30)		
Senior high school	11 (55)	10 (50)	9 (45)	4.40	0.62
University/college	4 (20)	7 (35)	4 (20)		
Marital status					
Single	16 (80)	19 (95)	19 (95)		
Married	2 (10)	0 (0)	0 (0)		
Divorced	1 (5)	1 (5)	1 (5)	6.33	0.39
Separated	1(5)	0 (0)	0 (0)		
Living condition					
Living alone	4 (20)	3 (15)	7 (35)		
Living with family	16 (80)	17 (85)	13 (65)	4.00	0.41
Duration of illness (years)	23.60 (10.14)	24.15 (8.94)	21.80 (10.71)	0.31	0.74
Chlorpromazine equivalents (mg/d)	444.25 (207.16)	463.50 (205.03)	451.60 (148.35)	0.05	0.95
CGI-S scores	3.01 (0.73)	3.10 (0.72)	3.20 (0.62)	0.42	0.69
BACS (global cognition) scores	−2.64 (1.27)	−3.10 (1.38)	−3.05 (1.41)	0.76	0.47

BMI: body mass index; BACS: Brief Assessment of Cognition in Schizophrenia; CGI-S: the Clinical Global Impression–Severity scale.

**Table 2 healthcare-13-03013-t002:** Results of descriptive and inferential statistics of the primary outcomes.

Outcome	Baduanjin (*n* = 20)	Brisk Walking (BW) (*n* = 20)	Health Education (HE) (*n* = 20)	Time	Group	Time × Group
	Mean (SD)	Mean (SD)	Mean (SD)	(*p*-Value; *F*; *η*^2^)	(*p*-Value; *F*; *η*^2^)	(*p*-Value; *F*; *η*^2^)
BACS- Verbal memory				0.77; 0.08; 0.01	**<0.01; 6.81; 0.19**	0.11; 2.28; 0.08
T1	−1.15 (0.68)	−1.31 (0.86)	−1.28 (0.73)			
T2	−0.98 (0.65) ^†^	−1.26 (0.86)	−1.25 (0.75)		**Baduanjin > HE**	
T3	−1.02 (0.75) ^†^	−1.25 (0.87)	−1.36 (0.78)		**Baduanjin > HE**	
BACS- Working memory				0.78; 0.07; <0.01	**<0.01; 5.57; 0.17**	0.73; 0.32; 0.01
T1	−1.35 (1.07)	−1.63 (1.15)	−1.54 (1.00)			
T2	−1.17 (1.01) ^†^	−1.39 (1.03) ^†^	−1.47 (0.95)		**BW > HE**	
T3	−1.17 (1.04) ^†^	−1.38 (1.05) ^†^	−1.52 (0.97)		**BW > HE**	
BACS- Motor speed				0.28; 1.17; 0.02	0.75; 0.29; 0.01	0.74; 0.31; 0.01
T1	−2.26 (0.83)	−2.39 (0.71)	−2.42 (0.70)			
T2	−2.18 (0.85)	−2.34 (0.71)	−2.40 (0.69)			
T3	−2.22 (0.83)	−2.37 (0.19)	−2.39 (0.74)			
BACS- Verbal fluency				0.32; 1.00; 0.02	0.11; 2.35; 0.08	0.91; 0.09; <0.01
T1	−0.92 (0.58)	−1.01 (0.61)	−0.96 (0.67)			
T2	−0.87 (0.62)	−0.91 (0.62)	−0.93 (0.66)			
T3	−0.89 (0.68)	−0.92 (0.56)	−0.98 (0.65)			
BACS-Attention and processing speed				0.78; 0.01; <0.01	**<0.01; 5.33; 0.16**	0.10; 2.42; 0.08
T1	−1.77 (0.99)	−1.85 (0.95)	−1.98 (1.02)			
T2	−1.52 (0.98) ^†^	−1.70 (0.89) ^†^	−1.92 (0.98)		**Baduanjin > HE**	
T3	−1.55 (1.04) ^†^	−1.63 (0.82) ^†^	−1.98 (1.06)		**Baduanjin > HE, BW > HE**	
BACS- Executive function				0.63; 0.23; <0.01	**0.04; 3.27; 0.11**	0.28; 1.31; 0.05
T1	−1.73 (1.17)	−1.88 (0.92)	−1.84 (1.09)			
T2	−1.55 (1.06)	−1.83 (1.04)	−1.85 (1.18)			
T3	−1.57 (1.08)	−1.77(0.92)	−1.98 (1.06)		**Baduanjin > HE**	
BACS- Global cognition				0.79; 0.07; <0.01	**<0.01; 14.29; 0.34**	0.06; 2.99; 0.10
T1	−2.64 (1.27)	−3.10 (1.38)	−3.05 (1.41)			
T2	−2.39 (1.20) ^†^	−2.90 (1.33) ^†^	−3.03 (1.36)		**Baduanjin > HE, BW > HE**	
T3	−2.44 (1.25) ^†^	−2.87 (1.28) ^†^	−3.14 (1.45)		**Baduanjin > HE, BW > HE**	

^†^ *p* < 0.01 indicates significant within-group difference. T1: Baseline; T2: post-intervention; T3: 4 weeks after the intervention. BACS: Brief Assessment of Cognition in Schizophrenia.

**Table 3 healthcare-13-03013-t003:** Results of descriptive and inferential statistics of the secondary outcomes.

Outcome	Baduanjin (*n* = 20)	Brisk Walking (BW) (*n* = 20)	Health Education (HE) (*n* = 20)	Time	Group	Time × Group
	Mean (SD)	Mean (SD)	Mean (SD)	(*p*-Value; *F*; *η*^2^)	(*p*-Value; *F*; *η*^2^)	(*p*-Value; *F*; *η*^2^)
6MWT				0.88; 0.02; 0.01	**<0.01; 8.17; 0.22**	0.26; 1.37; 0.05
T1	462.34 (40.26)	456.47 (39.90)	450.01 (37.49)			
T2	475.09 (44.28) ^†^	478.05 (39.22) ^†^	452.40 (36.44)		**BW > HE**	
T3	484.31 (47.45) ^†^	481.76 (39.44) ^†^	455.39 (40.04)		**Baduanjin > HE, BW > HE**	
30CST				0.55; 0.36; 0.01	**<0.01; 8.26; 0.23**	0.09; 2.46; 0.081
T1	16.45 (3.05)	17.05 (2.67)	16.40 (3.76)			
T2	17.90 (3.68) ^†^	18.45 (3.02) ^†^	16.95 (3.73)		**Baduanjin > HE, BW > HE**	
T3	18.15 (3.68) ^†^	18.40 (3.12) ^†^	16.75 (3.61)		**Baduanjin > HE, BW > HE**	
TUG				0.19; 1.78; 0.03	**<0.01; 13.79; 0.33**	0.14; 2.04; 0.07
T1	7.08 (0.94)	7.04 (1.22)	7.53 (1.09)			
T2	6.55 (0.88) ^†^	6.81 (1.14) ^†^	7.40 (1.13)		**Baduanjin > BW, Baduanjin > HE**	
T3	6.46 (0.91) ^†^	6.78 (1.07) ^†^	7.48 (1.01)		**Baduanjin > BW, Baduanjin > HE**	
TUG-manual				0.23; 1.48; 0.03	**<0.01; 11.68; 0.29**	0.34; 1.01; 0.04
T1	11.29 (1.63)	11.02 (1.57)	11.14 (1.66)			
T2	10.26 (1.05) ^†^	10.82 (1.55)	11.01 (1.56)		**Baduanjin > BW, Baduanjin > HE**	
T3	10.40 (1.02) ^†^	10.87 (1.54)	11.28 (1.61)		**Baduanjin > BW, Baduanjin > HE**	
TUG- cognitive				0.38; 0.79; 0.01	**<0.01; 14.63; 0.34**	0.14; 2.08; 0.07
T1	13.75 (1.91)	14.19 (2.07)	13.66 (2.30)			
T2	12.95 (2.09) ^†^	13.85 (1.94)	13.75 (2.23)		**Baduanjin > BW, Baduanjin > HE**	
T3	12.46 (2.03) ^†^	13.64 (1.83)	13.70 (2.39)		**Baduanjin > BW, Baduanjin > HE**	

^†^ *p* < 0.01 indicates significant within-group difference. T1: Baseline; T2: post-intervention; T3: 4 weeks after the intervention. 6MWT: Six-Minute Walk Test; 30CST: 30-s chair stand test; TUG: Timed Up-and-Go Test; TUG manual: the motor Timed Up-and-Go dual task; TUG-cognitive: the cognitive Timed Up-and-Go dual task.

## Data Availability

The data presented in this study are available on request from the corresponding author due to protect the privacy of the research participants.

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
