# Peer review of "Efficacy of Baduanjin Versus Brisk Walking on Cognitive and Physical Functions in Schizophrenia: A Three-Arm Randomized Controlled Trial"

_healthcare, 2025, doi:10.3390/healthcare13233013_

Round 1

Reviewer 1 Report

Comments and Suggestions for Authors

Dear Authors,

    In a small-sample randomized controlled trial (RCT), the authors explored and compared the effects of Baduanjin, brisk walking (BW), and health education (HE) on cognitive ability and physical function in patients with schizophrenia. From my perspective, the authors’ research design, structure, statistical methods, and writing are well-organized and standardized. This is particularly reflected in several merits in data processing and statistical analysis, such as the use of analysis of covariance (ANCOVA) to adjust for potential bias from baseline measurements, Bonferroni correction for multiple comparisons, and reporting effect sizes using eta squared (η²) from ANOVA. These strengths contribute to clear clinical and statistical interpretation. Nevertheless, several issues deserve attention.

  1. RCTs should adhere to the CONSORT (Consolidated Standards of Reporting Trials) statement, with the most recent version being CONSORT 2025, updated in 2025. The authors should provide the relevant checklist for reviewers.
  2. Regarding the published protocol followed by the intervention group (Reference 19), details of the intervention—such as the dates and setting of the trial—should be described.
  3. In the within-group multiple comparisons presented in Tables 2 and 3, comparisons between T2 and T3 appear to be lacking. Such comparisons would help determine whether the effects were maintained over time.
  4. Given the randomized group allocation, assessment of patients’ mental state may not be strictly necessary, though its inclusion could still be beneficial.

Author Response

Thank you for your positive evaluation of the study design and implementation. We have revised the manuscript according to your suggestions. Below is our point-by-point response.

Comment 1:
RCTs should adhere to the CONSORT (Consolidated Standards of Reporting Trials) statement, with the most recent version being CONSORT 2025, updated in 2025. The authors should provide the relevant checklist for reviewers.

Response 1:
We appreciate the reviewer’s comment. The CONSORT 2025 checklist has been provided.

Comment 2:
Regarding the published protocol followed by the intervention group (Reference 19), details of the intervention—such as the dates and setting of the trial—should be described.

Response 2:
We have added further details regarding the dates of the clinical trial in lines 125–127:
Recruitment, intervention delivery, and follow-up assessments were conducted between January 2024 and January 2025 at the psychiatric day care center of a medical center in Taiwan.”

Comment 3:
In the within-group multiple comparisons presented in Tables 2 and 3, comparisons between T2 and T3 appear to be lacking. Such comparisons would help determine whether the effects were maintained over time.

Response 3:

Thank you for your suggestion. Because the within-group comparisons between T2 and T3 for both the primary and secondary outcomes were not significant, no changes were made to Tables 2 and 3. However, we added clarifying statements in the Results section to enhance readers’ understanding.

In Section 3.2 Primary outcomes (lines 273–276), we added:
There were no significant differences in primary outcomes between the post-test and follow-up assessments, suggesting that the intervention effects were maintained during the follow-up period (Table 2).”
In Section 3.3 Secondary outcomes (lines 303–305), we added:
There were no significant differences in secondary outcomes between the post-test and follow-up assessments, indicating that the intervention effects were sustained over the follow-up period (Table 3).”

Comment 4:
Given the randomized group allocation, assessment of patients’ mental state may not be strictly necessary, though its inclusion could still be beneficial.

Response 4:

Thank you for your comment. We have added the following clarification to the limitations section to strengthen the quality of the manuscript(lines 439–444):

“…Although standardized symptom scales were not administered, overall clinical stability was monitored through the CGI-S, and the scores remained stable in both the Baduanjin and brisk walking groups. This stability suggests that the interventions were well tolerated and supports the safety of incorporating structured exercise into rehabilitation for individuals with schizophrenia...”

Reviewer 2 Report

Comments and Suggestions for Authors

Dear Authors,

Thank you for the opportunity to thoroughly review your manuscript. Your study is an excellent example of a rigorous three-arm RCT in clinical psychophysiology. It has great value because the effects of the Baduanjin and brisk walking (BW) interventions are compared not only with an active control (HE) but also during the crucial, often neglected, maintenance phase. The findings are clear and clinically meaningful.

My critique will focus on methodological, statistical, and discussion-related nuances that could enhance transparency and credibility, especially in the context of potential bias in RCTs.

Abstract

The abstract is nearly exemplary, correctly reporting Wilks’ Lambda (F-tests) and partial eta squared as measures of effect size.

Although you mention improvements in working memory and global cognitive function in the BW group, please consider adding a reference to the maintenance effect (specifically in Attention and Processing Speed) for BW in the abstract, as this is an important finding for that group.

In the Results section, please clarify that BW also produced significant benefits compared to HE (e.g., in working memory and global cognition). The current structure might imply that Baduanjin was the only effective intervention.

Also, please correct the list of physical measures in the Results section: the dual-task assessments (TUG and dual-task TUG tests) are currently listed twice.

Introduction

The introduction effectively situates the problem, emphasizing that antipsychotic medications have limited impact on negative symptoms and cognitive function. It clearly identifies the research gap—the lack of comparative RCTs (Baduanjin vs. BW) that include a maintenance phase.

The rationale for Baduanjin is strong, highlighting that it is easier to practice than Tai Chi and is a form of light-to-moderate intensity exercise. The hypothesis is clear: Baduanjin and BW would outperform HE, and Baduanjin and BW would show distinct effects on outcome measures.

Please mention that in schizophrenia research, the effects of exercise on psychiatric symptoms (e.g., positive/negative) have not been evaluated. This point appears later in the Limitations (Section 4.1), but including it in the Introduction would strengthen the rationale for including clinical symptom measures in future research.

Methods and Materials

The study was single-blinded, meaning that only outcome assessors were blinded, while participants and interventionists (occupational therapists, nurse) were aware of group allocation. In behavioral and psychological interventions, lack of participant blinding introduces a high risk of performance and expectancy bias. Participants in the Baduanjin and BW groups may have believed or expected that their intervention was more “therapeutic” than HE. The Discussion should include a more detailed reflection on this issue. Although it is mentioned in the Limitations, please add to the Discussion how the active control (HE: education + video watching) may have mitigated this bias, while acknowledging that it could not fully eliminate subjective expectations of better outcomes in the exercise groups.

The interventions (Baduanjin, BW, and HE) are precisely described, matched in duration (12 weeks, 3 sessions/week, 60 minutes each), and supervised. Although HE (health education and viewing non-exercise-related videos) serves as an active control that balances attention effects, the choice of content (sports games or sports-related variety shows) should be justified. Such content could have uncontrolled effects on motivation and engagement, potentially influencing cognitive outcomes indirectly. Please briefly comment on this in the Discussion.

The manuscript uses the terms motor dual-task of TUG (TUG-manual) and cognitive dual-task of TUG (TUG-cognitive). For full transparency, please briefly describe in the Methods (or Discussion) the specific cognitive task used in TUG-cognitive. The type of task (e.g., serial subtraction vs. categorization) substantially affects executive load and the comparability of results.

Baduanjin and BW showed comparable ratings of perceived exertion (RPE) (4.67 vs. 5.02), suggesting equivalent subjective load. Although Baduanjin is a mindful exercise, its intensity is sometimes questioned. Please strengthen the discussion that the comparable RPE values confirm that differences in outcomes were due to exercise type (mindful, balance-oriented) rather than intensity alone. This is crucial for clinical interpretation.

Results

Baduanjin was the only group showing significant within-group improvement from T1 to T3 (follow-up). This is the most important finding of your study and should be more strongly emphasized, as Baduanjin—unlike BW—maintained effects in this domain, which is critical for social functioning recovery.

Partial eta squared values are consistently reported, which is commendable. Global cognition and TUG demonstrate large effect sizes (per Cohen’s conventions). Please explicitly discuss these large values in the Discussion to strengthen the clinical relevance of Baduanjin as a modifying intervention.

Table 2 shows no significant effects for BACS-Motor speed. Please briefly discuss this “null result” in the Discussion. While BW is a motor exercise and Baduanjin requires precise movement, the lack of improvement in this domain (despite TUG gains) is intriguing and may suggest a specific neurological deficit that is not easily modified by general exercise.

Discussion and Limitations

The discussion of Baduanjin should more clearly separate potential mechanisms: Mindful Component (enhanced executive function, balance) vs. Aerobic Component (improved working memory). The superiority of Baduanjin over BW in dual-task TUG strongly suggests an advantage of the mindful/balance component (linked to prefrontal cortex and cerebellar activation) over a purely aerobic mechanism—this is a key translational insight from your study.

You effectively argue that the longer weekly training time for Baduanjin (180 minutes) and inclusion of a maintenance phase may explain the superior executive and dual-task outcomes compared with previous studies. Highlighting that verbal memory, processing speed, and executive function are critical predictors of return to work/school in schizophrenia underscores the clinical importance of Baduanjin.

The Limitations section rightly notes the lack of psychiatric symptom assessment. Although this is a limitation, please note that CGI-S (Clinical Global Impression–Severity) was monitored and remained stable in the Baduanjin and BW groups. This stability is an important positive finding, suggesting that the intervention is safe and does not exacerbate psychotic symptoms.

Adherence during the maintenance phase (4–5 sessions/week, ~20 min/session) was very high. Please emphasize that this strong adherence reflects good acceptability and feasibility of the intervention.

Conclusions

The conclusions are precise. However, it would be valuable to explicitly highlight that the superiority of Baduanjin in balance and dual-task performance has fundamental implications for daily functioning and safety among patients with schizophrenia—key aspects of their vocational and social rehabilitation.

English Language Evaluation

The English is academic, fluent, and accurate. Terminology is used consistently. The manuscript is very well written and does not require substantial editing—only minor proofreading before publication.

Author Response

Thank you for your valuable suggestions. We have revised the manuscript in accordance with your recommendations, as outlined below, and hope that these changes further improve the quality of the manuscript.

Comment 1:Abstract
The abstract is nearly exemplary, correctly reporting Wilks’ Lambda (F-tests) and partial eta squared as measures of effect size.
Although you mention improvements in working memory and global cognitive function in the BW group, please consider adding a reference to the maintenance effect (specifically in Attention and Processing Speed) for BW in the abstract, as this is an important finding for that group.
In the Results section, please clarify that BW also produced significant benefits compared to HE (e.g., in working memory and global cognition). The current structure might imply that Baduanjin was the only effective intervention.

Response 1:

Thank you for your comment. We have added the following statement to the abstract to highlight the effects of brisk walking (lines 47–49):

BW significantly enhanced the working memory and global cognition compared with HE, with additional improvements in attention and processing speed at follow-up.”

Comment 2:
Also, please correct the list of physical measures in the Results section: the dual-task assessments (TUG and dual-task TUG tests) are currently listed twice.

Response 2:

We revised the description of the physical function assessments in the abstract as follows(lines 43-45):

Physical outcomes included the Six-Minute Walk Test (6MWT), 30-Second Chair Stand Test (30CST), Timed Up-and-Go (TUG), motor dual-task TUG (TUGmanual), and cognitive dual-task TUG (TUGcognitive).

Comment 3:Introduction

The introduction effectively situates the problem, emphasizing that antipsychotic medications have limited impact on negative symptoms and cognitive function. It clearly identifies the research gap—the lack of comparative RCTs (Baduanjin vs. BW) that include a maintenance phase.
The rationale for Baduanjin is strong, highlighting that it is easier to practice than Tai Chi and is a form of light-to-moderate intensity exercise. The hypothesis is clear: Baduanjin and BW would outperform HE, and Baduanjin and BW would show distinct effects on outcome measures.
Please mention that in schizophrenia research, the effects of exercise on psychiatric symptoms (e.g., positive/negative) have not been evaluated. This point appears later in the Limitations (Section 4.1), but including it in the Introduction would strengthen the rationale for including clinical symptom measures in future research.

Response 3:

Thank you for your careful review of the introduction. We have added a statement addressing the limited evidence regarding the effects of Baduanjin on psychiatric symptoms to strengthen the manuscript (lines 80–81):

“... however, empirical studies examining its effects on psychiatric symptoms and functional outcomes in individuals with schizophrenia remain limited.”

Comment 4:Methods and Materials

The study was single-blinded, meaning that only outcome assessors were blinded, while participants and interventionists (occupational therapists, nurse) were aware of group allocation. In behavioral and psychological interventions, lack of participant blinding introduces a high risk of performance and expectancy bias. Participants in the Baduanjin and BW groups may have believed or expected that their intervention was more “therapeutic” than HE. The Discussion should include a more detailed reflection on this issue. Although it is mentioned in the Limitations, please add to the Discussion how the active control (HE: education + video watching) may have mitigated this bias, while acknowledging that it could not fully eliminate subjective expectations of better outcomes in the exercise groups.

Response 4:

Thank you for your suggestion. We have added the following content to the Discussion section (lines 433–438):

Although assessors were blinded, participants and interventionists knew their group assignments, which may have introduced performance or expectancy bias, a common limitation in behavioral interventions [50]. The active control condition of health education and video viewing helped minimize this bias by providing comparable attention, social interaction, and session time across groups; however, participants’ expectations of greater benefit in the exercise groups could not be completely eliminated.”

Comment 5:
The interventions (Baduanjin, BW, and HE) are precisely described, matched in duration (12 weeks, 3 sessions/week, 60 minutes each), and supervised. Although HE (health education and viewing non-exercise-related videos) serves as an active control that balances attention effects, the choice of content (sports games or sports-related variety shows) should be justified. Such content could have uncontrolled effects on motivation and engagement, potentially influencing cognitive outcomes indirectly. Please briefly comment on this in the Discussion.

Response 5:

Thank you for your suggestion. We have added the following explanation in the Methods section regarding the selection of video content (lines 193–197):

Sports games and sports-related variety programs were selected because they are engaging, easy to follow, and do not elicit physical exertion, thereby serving as an attention-matched and expectancy-balanced control condition consistent with recommendations for active controls in behavioral and exercise trials [28].”

We also added the following statement in the Discussion section (lines 435–439):

The active control condition of health education and video viewing helped minimize this bias by providing comparable attention, social interaction, and session time across groups; however, participants’ expectations of greater benefit in the exercise groups could not be completely eliminated.”

Comment 6:
The manuscript uses the terms motor dual-task of TUG (TUG-manual) and cognitive dual-task of TUG (TUG-cognitive). For full transparency, please briefly describe in the Methods (or Discussion) the specific cognitive task used in TUG-cognitive. The type of task (e.g., serial subtraction vs. categorization) substantially affects executive load and the comparability of results.

Response 6:

Thank you for your suggestion. We have revised the description of the TUG procedures in the Methods section as follows (lines 219-226):

The Timed Up and Go Test (TUG) [37], along with the motor dual-task TUG (TUGmanual) and the cognitive dual-task TUG (TUGcognitive), was administered to evaluate balance and dual-task performance [38]. In the TUGmanual, participants performed the TUG while holding a cup filled to nine-tenths with water [38]. In the TUGcognitive, a number between 80 and 99 was randomly selected, and participants completed the TUG while performing serial subtraction by three [38]. For all TUG conditions, lower completion times indicated better functional performance."

Comment 7:
Baduanjin and BW showed comparable ratings of perceived exertion (RPE) (4.67 vs. 5.02), suggesting equivalent subjective load. Although Baduanjin is a mindful exercise, its intensity is sometimes questioned. Please strengthen the discussion that the comparable RPE values confirm that differences in outcomes were due to exercise type (mindful, balance-oriented) rather than intensity alone. This is crucial for clinical interpretation.

Response 7:

Thank you for your suggestion. We have revised the Discussion section as follows(line 362-367):

In the present study, Baduanjin and brisk walking showed similar perceived exertion ratings, indicating comparable intensity. This supports the interpretation that the higher benefits of Baduanjin stem from its mindful and balance-focused elements rather than differences in exercise intensity. Evidence also suggests that Baduanjin improves executive control through mechanisms beyond aerobic load [41].”

Comment 8:Results
Baduanjin was the only group showing significant within-group improvement from T1 to T3 (follow-up). This is the most important finding of your study and should be more strongly emphasized, as Baduanjin—unlike BW—maintained effects in this domain, which is critical for social functioning recovery.

Response 8:

Thank you for your comment. We have revised the Discussion section as follows (lines 339–341):

The current findings show that Baduanjin was the only intervention to produce significant within-group improvements from baseline to follow-up and yielded higher gains in verbal memory compared with the health education group."

Comment 9:
Partial eta squared values are consistently reported, which is commendable. Global cognition and TUG demonstrate large effect sizes (per Cohen’s conventions). Please explicitly discuss these large values in the Discussion to strengthen the clinical relevance of Baduanjin as a modifying intervention.

Response 9:

Thank you for your comment. We have made the following revisions in the Discussion section:

Lines 393–394:
Our findings indicate that both Baduanjin and brisk walking produced significant and large improvements in global cognition compared with health education.”

Lines 419–420:
In our study, Baduanjin produced larger and sustained improvements in balance than both brisk walking and health education, with a large effect size.”

Comment 10:
Table 2 shows no significant effects for BACS-Motor speed. Please briefly discuss this “null result” in the Discussion. While BW is a motor exercise and Baduanjin requires precise movement, the lack of improvement in this domain (despite TUG gains) is intriguing and may suggest a specific neurological deficit that is not easily modified by general exercise.

Response 10:

Thank you for your comment. We made the following revisions in the Discussion section(lines 384-392):

Neither Baduanjin nor brisk walking produced significant changes in the BACS motor speed. Psychomotor slowing is common in schizophrenia and is closely related to functional performance [46], and recent studies likewise report the limited effects of exercise or cognitive training on this domain [23,47]. One possible explanation is that the BACS Motor Speed subtest requires participants to place coins into a bowl as quickly as possible, emphasizing rapid upper-limb movement [31], whereas Baduanjin and brisk walking do not specifically target upper-limb movement speed. Future research may examine whether activities involving fast upper-limb coordination, such as ball games, yield greater improvements on this measure.”

Comment 11:Discussion and Limitations

The discussion of Baduanjin should more clearly separate potential mechanisms: Mindful Component (enhanced executive function, balance) vs. Aerobic Component (improved working memory). The superiority of Baduanjin over BW in dual-task TUG strongly suggests an advantage of the mindful/balance component (linked to prefrontal cortex and cerebellar activation) over a purely aerobic mechanism—this is a key translational insight from your study.

Response 11:

Thank you for your suggestion. We revised the Discussion section as follows:

Lines 364–367:
This supports the interpretation that the higher benefits of Baduanjin stem from its mindful and balance-focused elements rather than differences in exercise intensity. Evidence also suggests that Baduanjin improves executive control through mechanisms beyond aerobic load [41].”

Lines 419–422:
In our study, Baduanjin produced higher and sustained improvements in balance than both brisk walking and health education, with a large effect size. This effect may be attributable to the emphasis on balance control in Baduanjin movements, including weight shifting and trunk rotation.”

Comment 12:

You effectively argue that the longer weekly training time for Baduanjin (180 minutes) and inclusion of a maintenance phase may explain the superior executive and dual-task outcomes compared with previous studies. Highlighting that verbal memory, processing speed, and executive function are critical predictors of return to work/school in schizophrenia underscores the clinical importance of Baduanjin.

The Limitations section rightly notes the lack of psychiatric symptom assessment. Although this is a limitation, please note that CGI-S (Clinical Global Impression–Severity) was monitored and remained stable in the Baduanjin and BW groups. This stability is an important positive finding, suggesting that the intervention is safe and does not exacerbate psychotic symptoms.

Response 12:

Thank you for your valuable comment. We made the following revision in the Discussion section (lines 440-444):

“…Although standardized symptom scales were not administered, overall clinical stability was monitored through the CGI-S, and the scores remained stable in both the Baduanjin and brisk walking groups. This stability suggests that the interventions were well tolerated and supports the safety of incorporating structured exercise into rehabilitation for individuals with schizophrenia.”

Comment 13:
Adherence during the maintenance phase (4–5 sessions/week, ~20 min/session) was very high. Please emphasize that this strong adherence reflects good acceptability and feasibility of the intervention.

Response 13:

Thank you for your comment. We made the following revision in the Discussion section (lines 403–408):

Participants reported practicing approximately four to five times per week for about 20 minutes per session, which aligns with prior findings and suggests that the interventions were acceptable and feasible for this population [16]. Such sustained engagement may have contributed to maintaining the intervention benefits and highlights the value of targeted follow-up strategies to support ongoing physical activity in individuals with schizophrenia.”

Comment 14:Conclusions
The conclusions are precise. However, it would be valuable to explicitly highlight that the superiority of Baduanjin in balance and dual-task performance has fundamental implications for daily functioning and safety among patients with schizophrenia—key aspects of their vocational and social rehabilitation.

Response 14:

Thank you for your comment. We made the following revision in the Conclusion section (lines 451–454):

Baduanjin exhibited superior effects on balance and dual-task performance compared with the other groups. These findings highlight the potential of Baduanjin as a promising form of aerobic exercise that may be integrated into psychiatric interventions for individuals with schizophrenia.”

Comment 15:English Language Evaluation

The English is academic, fluent, and accurate. Terminology is used consistently. The manuscript is very well written and does not require substantial editing—only minor proofreading before publication.

Response 15:
Thank you for your comment. We have sought MDPI’s language editing service to further improve the quality of the manuscript.